# The importance of A-site cation chemistry in superionic halide solid electrolytes

Kit Barker[1], Sarah L. McKinney[2,3,4], Raül Artal ●[5], Ricardo Jiménez[5], Nuria Tapia-Ruiz ●[2,3], Stephen J. Skinner ●[1], Ainara Aguadero ●[1,5] & Ieuan D. Seymour ●[1,6] ✉

Halide solid electrolytes do not currently display ionic conductivities suitable for high-power all-solid-state batteries. We explore the model system $A_2ZrCl_6$ (A = Li, Na, Cu, Ag) to understand the fundamental role that A-site chemistry plays on fast ion transport. Having synthesised the previously unknown $Ag_2ZrCl_6$ we reveal high room temperature ionic conductivities in $Cu_2ZrCl_6$ and $Ag_2ZrCl_6$ of $1 \times 10^{-2}$ and $4 \times 10^{-3}$ S cm$^{-1}$, respectively. We introduce the concept that there are inherent limits to ionic conductivity in solids, where the energy and number of transition states play pivotal roles. Transport that involves multiple coordination changes along the pathway suffer from an intrinsic minimum activation energy. At certain lattice sizes, the energies of different coordinations can become equivalent, leading to lower barriers when a pathway involves a single coordination change. Our models provide a deeper understanding into the optimisation and design criteria for halide superionic conductors.

Achieving fast ion mobility in a material at room temperature while maintaining structural and chemical integrity is an essential property in energy conversion and storage systems. While current Li-ion batteries contain a liquid electrolyte to achieve sufficient ionic conductivities of $10^{-3}$ - $10^{-2}$ S cm$^{-1}$ [1,2], the stability of electrolyte molecules imposes constraints on electrode compatibility and, subsequently, the energy density. Solid electrolytes are attractive owing to their potential affinity for alternative battery chemistries, such as a Li metal anode, possessing a remarkably high theoretical specific capacity (3860 mAh/g).

The materials suggested for use as solid-state electrolytes fall under one of four main categories: oxides, sulphides, halides and polymers (Supplementary Fig. 1). Halides have returned as a class of solid electrolyte materials proposed for all-solid-state batteries showing compatibility with high voltage cathodes[3–5]. Chlorides and fluorides show the best performance regarding electrochemical stability and are lighter than bromides and iodides. However, they display lower ionic conductivities[4,6,7]. By investigating features such as structure, cation size, covalency, thermodynamics and diffusion pathways,

we enhance our understanding of ion transport, which can be applied as materials design principles to achieve high ionic conductivities.

Halide materials and their associated ion transport properties have been studied for decades. Early research found many superionic iodides, such as the high-temperature phase of $\alpha$-AgI[8]. Materials falling into the class of "advanced superionic conductors" have the highest fastest ionic transport of materials known to date with conductivities above 0.1 S cm$^{-1}$ at room temperature[9]. Examples include $RbAg_4I_5$ and $Rb_4Cu_{16}I_7Cl_{13}$[10,11], where the interconnected, partially occupied face-sharing tetrahedral sites enable these materials to achieve ionic conductivities of 0.26 S cm$^{-1}$ and 0.21 S cm$^{-1}$, respectively[12,13]. To date, no pure chloride-based advanced superionic conductors have been discovered. Finding material for this class would be a monumental step towards achieving all-solid-state batteries with wide applications.

$Li_2ZrCl_6$ is an important candidate for use as a solid electrolyte. The use of $Zr^{4+}$ is attractive as efforts are being made to use cheaper, more abundant elements in the battery industry instead of rare earth or post-transition metal elements, such as Y and In in $Li_3YCl_6$ and $Li_3InCl_6$ (Supplementary Fig. 2). Kwak et al. demonstrated that $Li_2ZrCl_6$

[1]Department of Materials, Imperial College London, London, UK. [2]Department of Chemistry, Imperial College London, London, UK. [3]The Faraday Institution, Didcot, UK. [4]Department of Chemistry, Lancaster University, Lancaster, UK. [5]Instituto de Ciencia de Materiales de Madrid, CSIC, Madrid, Spain. [6]Advanced Centre for Energy and Sustainability (ACES), Department of Chemistry, University of Aberdeen, Aberdeen, UK. ✉e-mail: ieuan.seymour08@imperial.ac.uk

can exist in two polymorphs depending on the synthesis method[14]. The difference in magnitude in ionic conductivities for these two polymorphs is large ($10^{-4}$ vs $10^{-6}$ S cm$^{-1}$ at room temperature for the hexagonal close-packed (HCP) and cubic close-packed (CCP) structures, respectively).

Na$_2$ZrCl$_6$ has been proposed as a cheap Na halide solid electrolyte for all-solid-state Na batteries. As with so many of these halide systems, multiple crystal structures have been reported[15,16] (Supplementary note 1).

While Li-based argyrodite-type solid electrolytes are popular of late for use in all-solid-state batteries and Li-S batteries[17,18], Cu analogues such as Cu$_6$PS$_5$Br[19] were known long before. The ionic conductivity of this material was found to be $1.5 \times 10^{-5}$ S cm$^{-1}$ at room temperature with an activation energy of approximately 0.35 eV[19,20].

The compound Cu$_2$ZrCl$_6$ was reported in 2002[21], however, the electronic and electrolytic properties of this material have not been investigated. The structure of Cu$_2$ZrCl$_6$ was reported to fall in the same $P\bar{3}m1$ space group as the Li and Na systems.

In this work, we have synthesised Ag$_2$ZrCl$_6$, and we investigated the origins of rapid ion transport in the isostructural systems of Li$_2$ZrCl$_6$, Na$_2$ZrCl$_6$, Cu$_2$ZrCl$_6$, and Ag$_2$ZrCl$_6$. Through a combination of experimental methods and state-of-the-art atomistic modelling, we highlight the significant role that site preference and disorder play in influencing the conductivity of halide materials. These findings open up promising pathways for the future enhancement and design of halide solid-electrolyte systems.

## Results

### Structural and thermodynamic features of A$_2$ZrCl$_6$ (A = Li, Cu, Na, Ag)

Ball milling has been a widely used approach for synthesising alkali halide-type materials in the scientific community. Recent work on Li$_2$ZrCl$_6$ and Na$_2$ZrCl$_6$ are examples of where employing this technique provides a relatively simple and effective way of making materials with attractive properties[4,22]. Other halides containing monovalent ions have, however, historically not been synthesised using mechanochemical ball milling-based techniques. The softness or deformability of Cu$^+$ and Ag$^+$ halides would suggest that a mechanochemical synthesis strategy would be a practical way of obtaining entirely new compounds or those previously made via traditional melt-quench methods.

Li$_2$ZrCl$_6$ and Na$_2$ZrCl$_6$ were successfully synthesised in this study using the ball milling approach as reported in other works[14,16]. Cu$_2$ZrCl$_6$ was also synthesised in this work using a simple ball milling route and is isostructural with Cu$_2$ZrCl$_6$ produced by high-temperature synthesis, as shown below. Most importantly, the ball milling approach allowed us to synthesise the Ag analogue, Ag$_2$ZrCl$_6$, which, to the best of our knowledge, has not been reported.

Supplementary Fig. 3 shows the Rietveld refinements of the four synthesised compounds using structures based on the $P\bar{3}m1$ space group. The refinement for Na$_2$ZrCl$_6$ utilised a minor $P2_1/n$ phase[15] to achieve the best fit.

Li$_2$ZrCl$_6$, Na$_2$ZrCl$_6$, Cu$_2$ZrCl$_6$ and Ag$_2$ZrCl$_6$ share many crystallographic features. The space group $P\bar{3}m1$ applies to all 4 compounds, sharing the same HCP chloride sublattice. Zr ions occupy octahedral sites on the 1a, 1b and 2d Wyckoff positions with varying degrees of site occupancy disorder (Fig. 1a). Li, Na and Ag ions occupy the octahedral 6g and 6h Wyckoff positions (Fig. 1b) while Cu occupies the tetrahedral 6i Wyckoff sites (Fig. 1c). Li$^+$ has been found in both octahedral and tetrahedral configuration in halide-based materials, depending on the size of the lattice and presence of neighbouring vacant sites. Although Li$^+$ and Cu$^+$ have similar ionic radii, additional mixing between the 3d$^{10}$ valence orbitals in Cu$^+$ with the higher energy 4s and 4p states leads to an additional stabilisation of the tetrahedral configuration, as shown in previous work[23,24]. A refinement was conducted for Cu$_2$ZrCl$_6$ where the

Cu$^+$ species were located on octahedral sites. This refinement led to a poor fit with ($\chi^2$ and R$_{wp}$ values of 17.90 and 7.85, respectively). In Li$_2$ZrCl$_6$, Na$_2$ZrCl$_6$ and Ag$_2$ZrCl$_6$, the A-site cations are octahedrally coordinated, sharing the edges of the polyhedra in the $ab$-plane while sharing faces in the $c$-direction.

Stoichiometric unit cells were designed for subsequent computational analysis. The DFT ground state structures for A$_2$ZrCl$_6$ were obtained by calculating the formation energies for the reaction:

$$\text{ZrCl}_4 + 2\,\text{ACl} \longrightarrow \text{A}_2\text{ZrCl}_6 \tag{1}$$

The lowest energy (ground state) configuration of Li$_2$ZrCl$_6$ (Fig. 1d) is found when all Li and Zr cations occupy the octahedral sites in one layer. The difference between the ground state and the highest energy configuration in Li$_2$ZrCl$_6$ is 0.048 eV/atom, suggesting that many configurations are accessible at room temperature (thermal energy at room temperature is approximately 0.025 eV).

Na$_2$ZrCl$_6$ and Ag$_2$ZrCl$_6$ are found to have the same ground state configuration (Fig. 1d). Zr is distributed across the two layers while the A-site cations are distributed evenly across the two layers, sharing edges with Zr, but not themselves. Figure 1e shows that Na$_2$ZrCl$_6$ has a more negative formation energy compared to Li$_2$ZrCl$_6$. Figure 1e also shows that Na$_2$ZrCl$_6$ has significantly more negative formation energy, with a larger gap between the lowest energy configuration and higher energy configurations than Ag$_2$ZrCl$_6$; there is more of an energy penalty moving from one configuration to another in Na$_2$ZrCl$_6$ than Ag$_2$ZrCl$_6$. Ag$_2$ZrCl$_6$ displays very limited chemical stability with only 12/192 configurations having a negative formation energy; furthermore, the difference between the ground state and highest energy configurations is only 0.033 eV/atom suggesting many configurations might be accessible at room temperature via entropic stabilisation.

For Cu$_2$ZrCl$_6$, many low energy configurations displayed a shift in position from tetrahedral to trigonal planar sites between the initial and final structures, respectively, similar to that seen via XRD at higher temperatures[21]. Cu atoms were also relaxed at the octahedral sites occupied by the other systems. All the Cu atoms fell into nearby trigonal planar or tetrahedral sites indicating that Cu atoms are inherently unstable in octahedral sites. The difference between the ground state and the highest energy configuration that we simulated is 0.133 eV/atom, which is not accessible at room temperature. Only 4 of the configurations where Zr is all in one layer are found to be stable, supporting the diffraction data. Figure 1e shows that in Cu$_2$ZrCl$_6$, many configurations are close in energy allowing facile movement between them.

Interestingly, when Li atoms were placed in the tetrahedral positions of the Cu$_2$ZrCl$_6$ ground state, the energy difference between this structure and the ground state was found to be 0.027 eV. Having interstitial sites relatively close in energy to the ground state can provide low-energy transition states for long-range diffusion. Upon relaxing Na into the tetrahedral structure, the difference in energy from the ground state was 0.054 eV/atom. This larger energy penalty suggests that pathways involving a tetrahedral intermediate will be less accessible. The tetrahedral configuration of Ag$_2$ZrCl$_6$ was calculated to be only 0.008 eV/atom higher than the ground state configuration. This vanishingly small energy difference shows that while the system has a small degree of chemical stability, the potential energy surface is smooth, providing low barriers for facile Ag$^+$ transport between configurations.

### A-site dependence of A$^+$ ionic conductivity

Nyquist plots for the different compounds at room temperature can be seen in Supplementary Fig. 4. For Li$_2$ZrCl$_6$ and Na$_2$ZrCl$_6$, one semicircle is present. The capacitance calculated via fitting is on the order of $10^{-11}$ F. This would suggest that the conductivity calculated reflects that of the total conductivity[25]. The conductivities measured of $5 \times 10^{-4}$ and $9 \times 10^{-6}$ S cm$^{-1}$ for Li$_2$ZrCl$_6$ and Na$_2$ZrCl$_6$, respectively, in Fig. 2a are

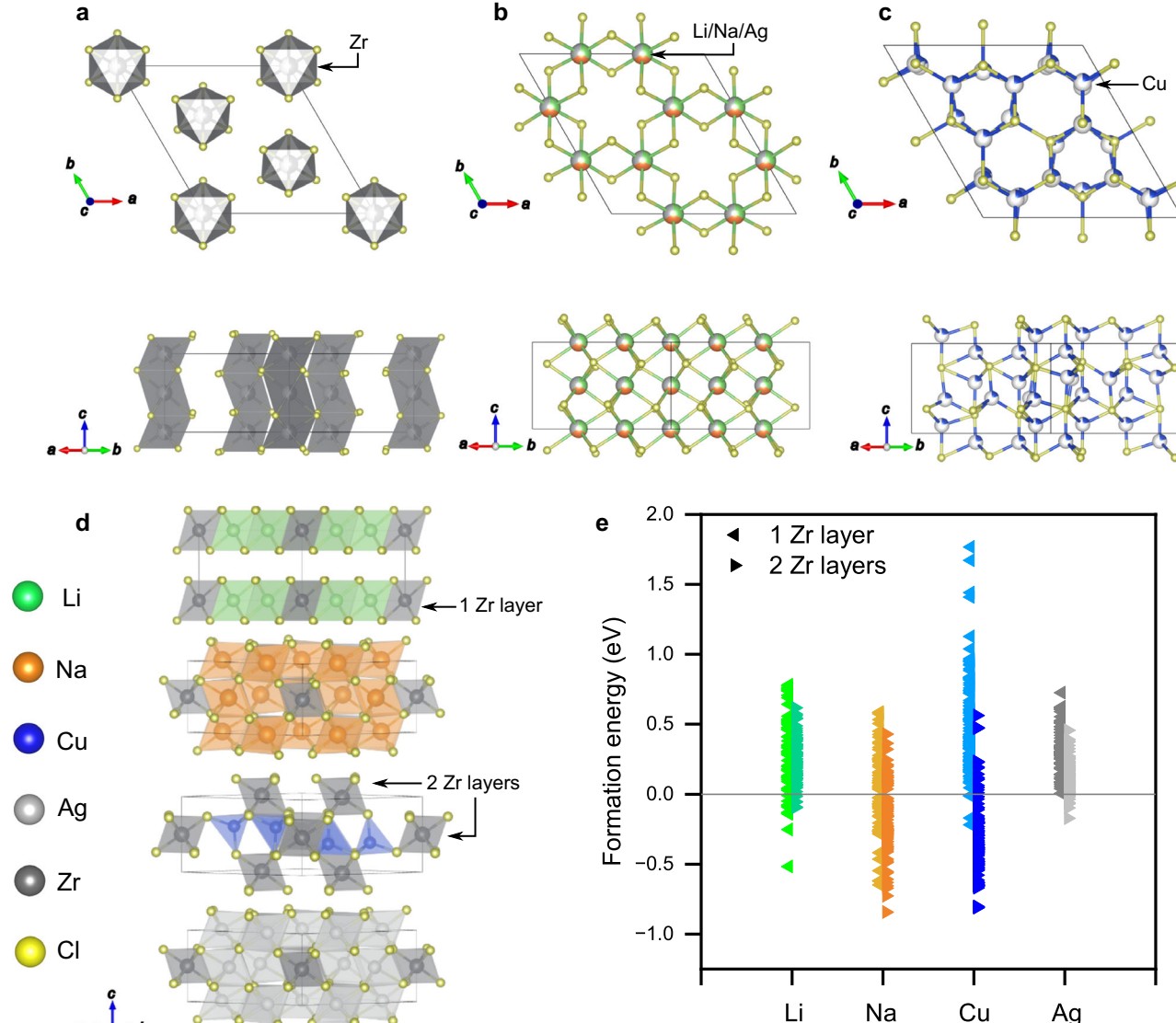

**Fig. 1 | Structural models of A$_2$ZrCl$_6$ materials.** Structural models of A$_2$ZrCl$_6$ found via XRD showing: (**a**) The disordered ZrCl$_6$ sublattice. **b** The octahedral positions of Li, Na and Ag. **c** The disordered tetrahedral positions of Cu. **d** The lowest energy configurations of A$_2$ZrCl$_6$ were found via DFT. **e** The DFT formation energies of different configurations in A$_2$ZrCl$_6$ with respect to the reagents used (2ACl + ZrCl$_4$).

similar to those reported in other work[14,16,26]. No EIS measurements have been conducted on Cu$_2$ZrCl$_6$ and Ag$_2$ZrCl$_6$ prior to this investigation. A semicircle is not observed in the Nyquist plots for Cu$_2$ZrCl$_6$ and Ag$_2$ZrCl$_6$. The x-axis intercept is taken to be the maximum total resistance for which the conductivity is calculated[27,28]. The values of $1 \times 10^{-2}$ S cm$^{-1}$ and $4 \times 10^{-3}$ S cm$^{-1}$ for Cu$_2$ZrCl$_6$ and Ag$_2$ZrCl$_6$, respectively, in Fig. 2a reveal that these are two excellent ion conductors. To the best of our knowledge, Cu$_2$ZrCl$_6$ exhibits the highest ionic conductivity of a crystalline chloride-type solid electrolyte to date. This suggests that the tetrahedral coordination in Cu$_2$ZrCl$_6$ is most favourable for ion transport within the ZrCl$_6$ host lattice. To test whether the trend in conductivity could be rationalised solely based on the covalent nature of the mobile ions within their respective structures, Bader charge analysis was used (Supplementary Fig. 5). Bader charge analysis demonstrated that the covalency of ions from most to least covalent was: Ag$^+$ > Cu$^+$ > Na$^+$ > Li$^+$, whereas the measured conductivity followed the trend: Cu$^+$ > Ag$^+$ > Li$^+$ >> Na$^+$, which suggests that additional structural factors have a critical impact on the conductivity.

Experimentally calculated activation energies for A$^+$ conductivity can be seen in Fig. 2(b). Li$_2$ZrCl$_6$ and Na$_2$ZrCl$_6$ display activation energies of 0.30 eV and 0.40 eV, respectively, similar to values reported in literature[14,16]. The values observed for Cu$_2$ZrCl$_6$ and Ag$_2$ZrCl$_6$ are 0.19 eV and 0.24 eV, respectively. The ionic transport properties of Cu$_2$ZrCl$_6$ and Ag$_2$ZrCl$_6$ have not been investigated previously to the best of our knowledge. The activation energies and pre-exponential factors show a strong Meyer-Neldel relationship where the pre-exponential factor decreases as the conductivity increases (Supplementary Fig. 4e). These results demonstrate that both Cu$_2$ZrCl$_6$ and Ag$_2$ZrCl$_6$ are exceptional ion conductors and could have application in low voltage, fast charging cells.

Ab initio molecular dynamics (AIMD) were performed to provide insight into the mobility of the A$^+$ cations in each of the structures (Supplementary Fig. 6). Anisotropic diffusion is observed for all 4 systems. The favoured direction is consistent with the pathway that involves a single coordination change. For A = Li$^+$, Na$^+$ and Ag$^+$, this is between octahedral (oct) sites through a 3-coordinate trigonal planar

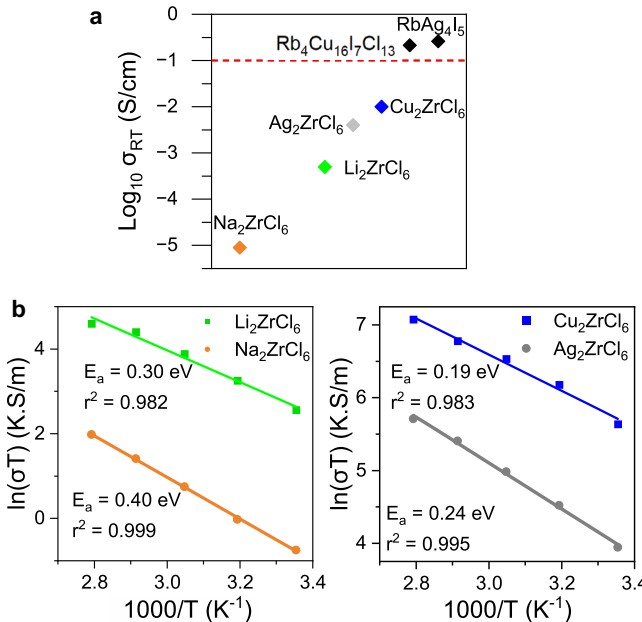

**Fig. 2 | Ionic conductivities of $A_2ZrCl_6$ materials. a** Room temperature ionic conductivities of $A_2ZrCl_6$ and the advanced superionic conductors $RbAg_4I_5$ and $Rb_4Cu_{16}I_7Cl_{13}$[12,13]. The dashed red line shows the threshold for `advanced superionic' conductivity of 0.1 S/cm. **b** Arrhenius plots for $A_2ZrCl_6$ and their associated activation energies calculated via EIS.

(trig) site along the $c$-axis. For $Cu^+$ ions in tetrahedral (tet) sites, a tet-trig-tet pathway in the $ab$-plane displays the most mobility.

Supplementary Fig. 6e shows the calculated activation energies for diffusion for $A^+$ ions in $A_2ZrCl_6$. $Na_2ZrCl_6$ has the largest activation barrier of 0.61 eV, which is higher than our experimentally observed value of 0.40 eV as well as values reported in literature[16]. It is expected that the ordered structures used in our AIMD calculations show lower conductivities than experimental data due to the conductivity-enhancing disorder during high energy milling[16,29]. $Li^+$ ions display an activation energy of 0.36 eV which is in excellent agreement with the findings of Wang et al.[30] and slightly higher than our experimentally determined value of 0.30 eV. The decrease in activation energy from Na to Li can be attributed to the impact of site preferences which are discussed later. $Cu_2ZrCl_6$ has been calculated to have a very low activation energy of 0.21 eV which is in agreement with the experimentally calculated value. Ag ions in $Ag_2ZrCl_6$ show a distinctly low activation energy for the diffusion of 0.11 eV, which is comparable to that of high-temperature $\alpha$-AgI, suggesting that the energies of different configurations and surrounding sites are very close to each other.

## Activation barriers and pathways of $A^+$ cations

AIMD simulations demonstrated that the size of the activation energy for $A^+$ ion transport in the $A_2ZrCl_6$ system was heavily influenced by the nature of the $A^+$ ion. To gain a deeper understanding of the atomic scale processes, we used a transition state searching (TSS) method to specifically pinpoint the unknown $A^+$ hopping processes. Due to the structures investigated in this work being inherently vacancy-rich, kinetically resolved barriers are used to reduce the effect that initial

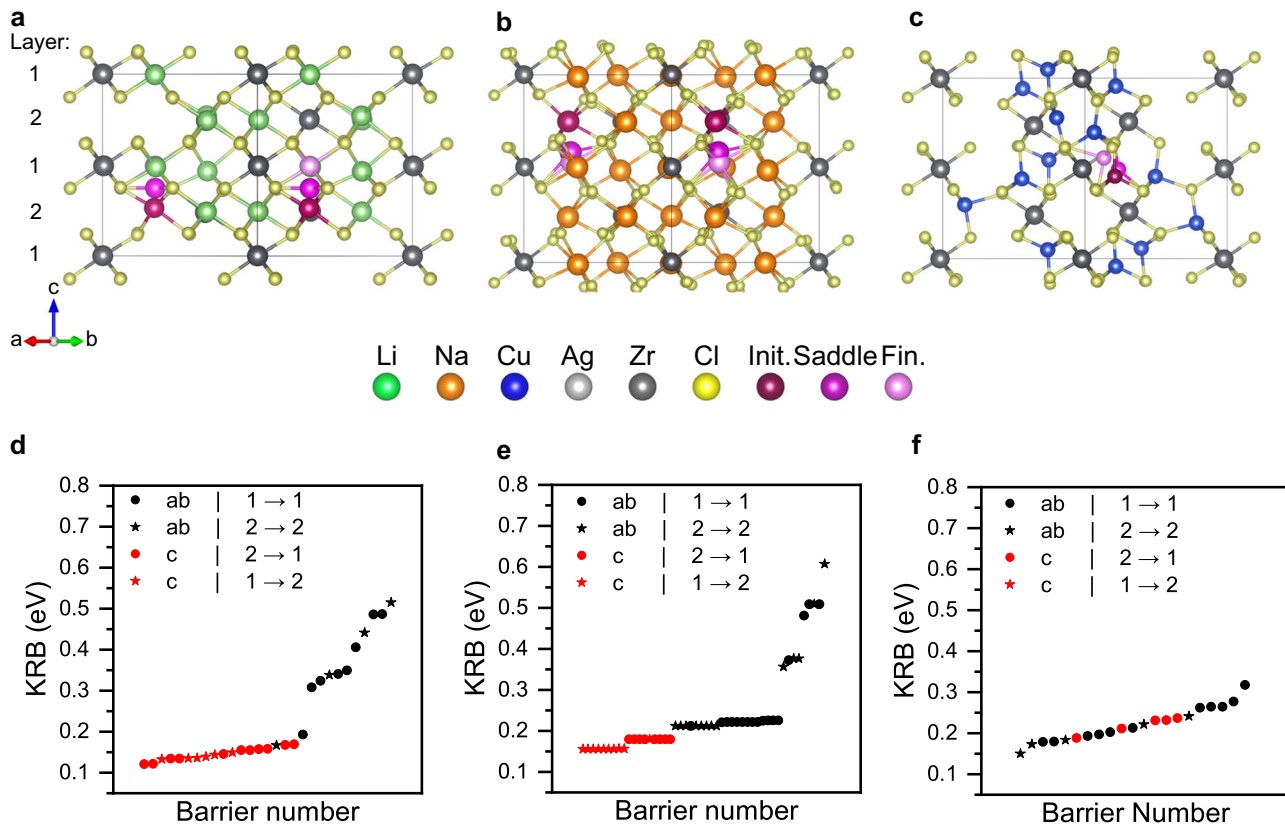

**Fig. 3 | Transition state searching activation energies for $A_2ZrCl_6$ materials.** Schematic representations of kinetically resolved activation barriers (KRB) were found for (**a**) $Li_2ZrCl_6$, (**b**) $Na_2ZrCl_6$ and (**c**) $Cu_2ZrCl_6$. Crimson, purple and pink balls represent the initial, saddle and final positions along their respective pathways. Activation energies of the unique mechanisms found via TSS calculations at 300 K in (**d**) $Li_2ZrCl_6$, (**e**) $Na_2ZrCl_6$ and (**f**) $Cu_2ZrCl_6$. Diffusion pathways are categorised by whether the diffusion is in the $ab$ or $c$-direction and whether diffusion occurs in layers with 1 or 2 Zr.

and final states have on calculated activation energies[31]. These are shown in Fig. 3. The energies of the initial (reactant), saddle and final (product) states are shown in Supplementary Fig. 7 to paint the full thermodynamic picture.

Example transport mechanisms of $Li_2ZrCl_6$, $Na_2ZrCl_6$ and $Cu_2ZrCl_6$ are shown in Fig. 3a−c, respectively. A $Li^+$ ion in $Li_2ZrCl_6$ (Fig. 3a) can be seen hopping from an octahedral site to an adjacent octahedral site along the $c$-axis via a trigonal planar transition state. This is one example of the barriers that were found via the TSS approach, demonstrating an excellent way to find and visualise transport mechanisms in a system while simultaneously mapping the energy landscape. Figure 3d shows the kinetically resolved barriers of Li diffusion processes found in $Li_2ZrCl_6$. There is a clear correlation between the direction in which the ion is moving and the height of the barrier. In agreement with our MD results, ion transport is favoured along the $c$-direction. When moving in the $c$-direction, the number of Zr in the initial layer vs the final layer has minimal effect on the barrier height i.e., Li going from a layer with 1 Zr to a layer with 2 Zr is thermodynamically as favourable as the reverse direction. While the number of Zr in each layer has little effect on transport in the $c$-direction, transport in the $ab$-plane is limited to isolated pathways in layers where Zr occupancy is high (Supplementary Fig. 8). The transition states for all of these barriers were found to have either trigonal planar or distorted tetrahedral coordination, highlighting the importance of lowering the energies of these sites to facilitate fast Li conduction in halide systems.

A similar mechanism can be seen for $Na_2ZrCl_6$ (Fig. 3b) that reflects the barriers found for transport in the $c$-direction. Similarly to $Li_2ZrCl_6$, 1-dimensional transport is favoured in the $c$-direction via a trigonal planar transition state, with an increase in activation energy compared to the Li system. Transport in the $ab$-plane is slightly more complex in $Na_2ZrCl_6$. Transition states are observed to have trigonal planar or highly distorted tetrahedral coordination, suggesting that the energy landscape is rough. Figure 3e shows that transport in the ab-plane has a larger associated activation energy than the $c$-direction, in agreement with our MD calculations.

Figure 3c shows one pathway within $Cu_2ZrCl_6$ found via TSS. An initial tetrahedral site can be seen travelling through a trigonal planar intermediate before settling in an adjacent tetrahedral position. Two transition states were observed in the barriers found via TSS: a trigonal planar intermediate and a linear one suggesting that Cu ions travel through both the faces and edges of their respective tetrahedra. Multiple barriers resulted in the mobile Cu ion moving to a trigonal planar product state, again demonstrating that the trigonal planar sites in $Cu_2ZrCl_6$ are indeed low in energy. The saddle point is a distorted octahedral site, as any other pathway would involve a site face sharing with Zr. The barriers displayed in Fig. 3f show that diffusion is more isotropic in $Cu_2ZrCl_6$ compared to $Li_2ZrCl_6$ and $Na_2ZrCl_6$. The barriers for $Cu_2ZrCl_6$ obtained via TSS are slightly higher than the activation energy calculated via MD, suggesting that the mechanism may be a cooperative process rather than a simple vacancy-mediated one. TSS was attempted for the $Ag_2ZrCl_6$ system, but the barriers were found to be so small (i.e., a flat energy landscape) that the saddle point searching methods used in the TSS process were unable to converge at a simulation temperature of 300 K. We, therefore, rely on the MD results of $Ag_2ZrCl_6$ to understand the diffusion mechanism.

### Coordination and transport limitations
Our TSS data has revealed that activation energies in $A_2ZrCl_6$ are influenced by A-site cation coordination and type. In the $ab$-plane, $Li_2ZrCl_6$, $Na_2ZrCl_6$, and $Ag_2ZrCl_6$ show ion hops between octahedral and tetrahedral sites, with a trigonal planar site as the transition state Fig. 4a. Along the $c$-axis, diffusion involves hops between octahedral sites through a trigonal planar site. $Cu_2ZrCl_6$ exhibits hopping along tetrahedral-trigonal planar-octahedral pathways, with direct hops

between tetrahedral sites via edges also observed. The activation energy depends on the relative energy of the transition states, which is influenced by the nearest neighbour Cl-Cl distance of the polyhedra. We examined the variation in site energy for model LiCl, NaCl, CuCl, and AgCl HCP systems as a function of unit cell volume, to understand the relative energies of sites in these systems.

A schematic representation of the variation in cell energy as a function of Cl-Cl distance can be seen in Fig. 4b. The raw plots for each of the structures can be seen in Supplementary Fig. 9. The calculations show that in different coordinations have different minimum energies at specific HCP ACl cell sizes. In LiCl, NaCl and AgCl, at short Cl-Cl distances, the octahedral coordination is lowest in energy (Fig. 4c). Conversely, at longer bond distances, the tetrahedral coordination is lowest in energy with other coordinations slightly higher in energy (local minima). CuCl, on the other hand, shows the lowest energy octahedral coordination only at very short Cl-Cl distances before the tetrahedral coordination becomes the ground state. Interestingly, at longer Cl-Cl distances the trigonal and linear coordinations are lowest in energy.

For diffusion in the $ab$-plane of octahedral $A_2ZrCl_6$ systems, A-site cations must diffuse through sequential octahedral-trigonal planar-tetrahedral configurations, in which the Cl-Cl distance is determined by the $A_2ZrCl_6$ lattice size. The size of the barrier is, therefore, strongly dependent on the maximum energy difference between the octahedral, trigonal and tetrahedral sites. The maximum energy difference for each ACl system is plotted in Fig. 4d.

At small Cl-Cl distances ($X_{Cl-Cl} < i$), the barrier is dictated by the energy difference of the octahedral (lowest E) and trigonal planar (highest E) sites. At intermediate distances (point i to point ii), the barrier is dictated by the energy difference between the tetrahedral (lowest E) and trigonal planar site (highest E). At long distances (point ii to point iii), the barrier is dictated by the difference between the tetrahedral (lowest E) and octahedral (highest E) sites. At the longest distance ($X_{Cl-Cl} > iii$), the barrier is dictated by the difference between trigonal planar (lowest E) and octahedral (highest E) sites. The shape of the curves is consistent with previous work by Wang et al.[32].

For all structures, the minimum energy point occurs at cell lengths where the 3-coordinate $A_{trig,ab}$ site is the same energy as the 6-coordinate $A_{oct}$ site (point ii), as shown in Fig. 4b, d. At this point, the 4-coordinate $A_{tet}$ site is the lowest energy. From Fig. 4b, c, as the minimum energy bond length increases as the coordination decreases, there is no point where the $A_{oct}(6)$- $A_{trig,ab}(3)$ - $A_{tet}(4)$ sites adopt the same energy (i.e., i, ii and iii do not cross at a single point). Importantly, this results in a fundamental, finite lower bound for the activation energy of diffusion within the $ab$-plane of $A_2ZrCl_6$ materials.

In contrast to $ab$-plane diffusion in the HCP structure, for $c$-axis diffusion in $A_2ZrCl_6$, only A hops between $A_{oct}$ and $A_{trig,c}$ site are required, and so if a material with the minimum energy Cl-Cl distance is selected (point ii), the $A_{oct}$ and $A_{trig}$ sites have equivalent energies, and there is no lower bound for the activation energy (Fig. 4e). This is consistent with the very small $c$-axis activation energies for $A_2ZrCl_6$ systems, even though there is a large change in the A coordination from 6 to 3 along the diffusion pathway.

For the CuCl system, the relative energy of the sites is $E(A_{tet})$ < $E(A_{trig})$ < $E(A_{oct})$. The $A_{oct}$ site, therefore, serves as the transition-state along the $A_{tet}$- $A_{trig}$- $A_{oct}$- $A_{trig}$- $A_{tet}$ pathway. These pathways are observed for $Cu_2ZrCl_6$. From TSS simulations, direct hops between tetrahedral sites are also observed, through a linear edge $A_{lin}$. At long Cl-Cl distances (4.14 Å), this pathway becomes favourable and free from an intrinsic barrier (Supplementary Fig. 10).

## Discussion
The model developed in this work provides a fundamental understanding of the role of coordination changes and A-site cation species on the activation energy for cation diffusion on HCP halide materials, which has so far been lacking.

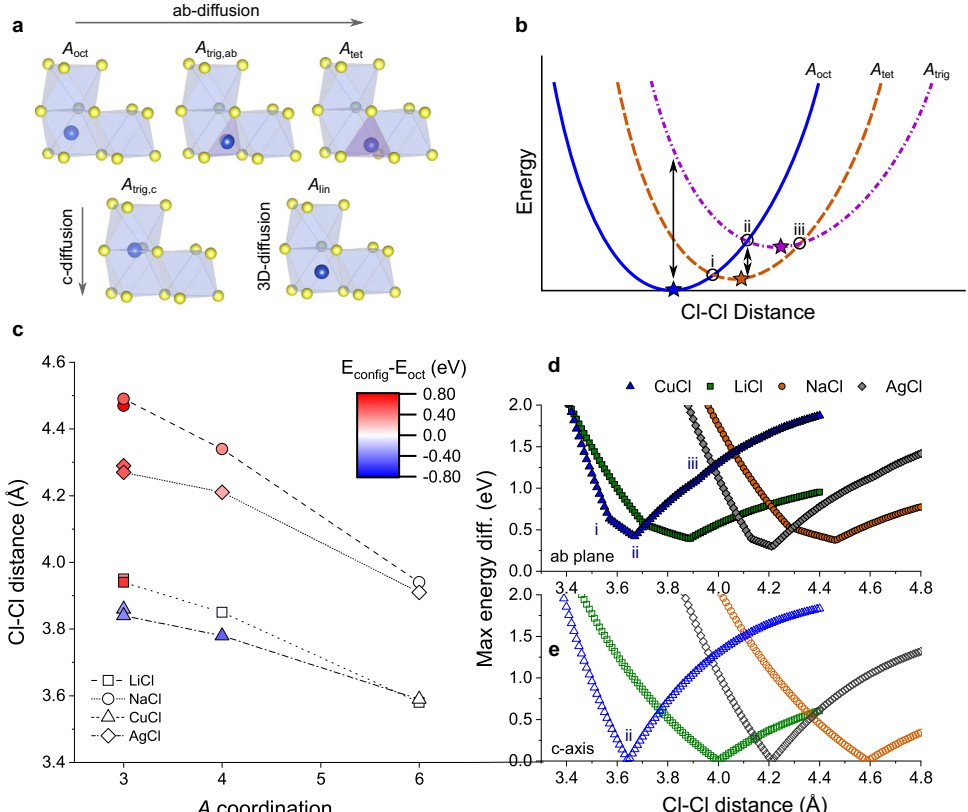

**Fig. 4 | Energy volume relationship for A₂ZrCl₆ activation energies. a** A schematic of different A cation configurations in HCP ACl structures, showing 6-coordinate octahedral ($A_{oct}$), 4-coordinate tetrahedral ($A_{tet}$), and 3-coordinate trigonal ($A_{trig,ab}$ and $A_{trig,c}$) configurations. A and Cl sites are labelled blue and yellow, respectively, and adjacent A-site cations are omitted for clarity. **b** Schematic diagram of energy variation of different cation sites ($A_{oct}$, $A_{tet}$, $A_{trig}$) and as a function of Cl-Cl distance (unit cell volume) in undistorted HCP ACl structures. The linear configuration, $A_{lin}$, has been omitted for clarity. The minimum energy for each A-site type is shown with a star. The Cl-Cl distances where the $A_{oct}/A_{tet}$, $A_{oct}/$ $A_{trig}$, and $A_{tet}/A_{trig}$ curves have equal energy are labelled as i, ii and iii, respectively. Arrows indicate the maximum energy difference between A-site types at different Cl-Cl distances. **c** Plot of Cl-Cl length at the minimum energy point vs A-site coordination ($A_{oct}$(6), $A_{tet}$(4), $A_{trig}$(3)) for HCP LiCl, NaCl, CuCl and AgCl structures. The colour scale shows the energy of the configurations relative to the octahedral site. Lines are included to guide the eye. The plot of maximum energy difference per formula unit between (**d**) $A_{oct}$ - $A_{tet}$ - $A_{trig,ab}$ sites and (**e**) $A_{oct}$ - $A_{trig,c}$, to model *ab*-plane and *c*-axis diffusion in HCP ACl structures, respectively. Points i–iii from (**b**) are labelled for the CuCl (blue) curves.

In Li₂ZrCl₆, Na₂ZrCl₆, and Ag₂ZrCl₆, *ab*-plane conduction primarily involves $A_{oct}$ (min)-$A_{trig}$ (saddle)-$A_{tet}$ (min) hops, imposing a lower bound on activation energy due to two coordination changes (6-3-4). Ag₂ZrCl₆ exhibits faster ionic conduction than Na₂ZrCl₆, attributed to a smaller energy difference between $A_{oct}$ and $A_{trig,ab}$ sites, despite the larger ionic radius of Ag. Li₂ZrCl₆ also shows a small energy difference but incurs additional penalties from different Li orderings (Fig. 1e). For *c*-axis conduction in these materials, face-sharing $A_{oct}$-$A_{trig,c}$ sites, changing coordination once along the pathway, facilitate fast diffusion despite a larger coordination change (6-3-6).

For the Cu₂ZrCl₆ system, the preference for $A_{tet}$ coordination leads to very high conduction in the *ab*-plane. For a $A_{oct}$- $A_{trig,ab}$- $A_{tet}$- $A_{trig,ab}$- $A_{oct}$ pathway, the lowest intrinsic barriers for any system occur when the energy of $E(A_{oct})= E(A_{trig,ab})$, which occurs near the point where $A_{tet}$ is the global minimum. This suggests that the tetrahedral Cu configuration in Cu₂ZrCl₆ is close to the optimum for conduction. In Cu₂ZrCl₆, additional pathways also exist involving direct $A_{tet}$- $A_{lin}$- $A_{tet}$ hops for which an intrinsic minimum barrier does not exist.

This result suggests that a possible avenue to improve the *ab*-conductivity in Li_yMCl₆ and Na_yMCl₆ (M = transition metal or lanthanide) systems is to push the material towards the optimal tetrahedral cation configuration. These systems will, however, still suffer from an intrinsic activation barrier due to multiple coordination changes along a pathway.

A promising strategy is to look for new families of materials in which single coordination changes are maintained throughout the diffusion pathway. These are facilitated by face-sharing polyhedra. Extremely fast ionic conductivity has been observed in RbAg₄I₅ systems, which involve Ag hops along face-sharing $A_{tet}$ (4)- $A_{trig}$ (3)- $A_{tet}$(4) pathways, that only involve a single change in coordination[33]. Analogous behaviour has also been observed in oxide systems, such as the P2 layered Na_yMO₂ systems involving single coordination change hops between face sharing 6 coordinate trigonal prismatic sites, through a 4-coordinate square planar site[34], and the TiNb₂O₇ system in which single coordinate change Li hops occur between 5-coordinate square pyramidal sites and 4-coordinate square planar sites[35]. The results also suggest that another strategy for finding Li superionic conductors is to search for analogous Cu⁺-based systems. Many may have favourable, unconsidered crystal structures where substitution for Li is possible.

Overall, our study makes use of a model system, A₂ZrCl₆, to explore ionic conductivity in halide-type solid electrolytes. By successfully synthesising Ag₂ZrCl₆, we unveil exceptional ion-conducting properties in both Cu₂ZrCl₆ and Ag₂ZrCl₆, closing the gap to achieving chloride-type advanced superionic conductors.

Through our comprehensive investigation, employing first-principles calculations and single-ended transition state searching, we discern mechanistic and energetic differences in the transport properties of these compounds based on the A-site element. Unusual

linear transition states allow for rapid diffusion in $Cu_2ZrCl_6$, while high energy face-sharing sites and intrinsically limited oct-trig-tet configurations limit diffusion in $Li_2ZrCl_6$. We introduce the concept that intrinsic limits to ionic conductivity in solids arise from a combination of chemical and structural factors, wherein the stability and number of transition states of the mobile species play crucial roles.

Our models significantly contribute to a deeper understanding of the optimisation and design criteria for halide superionic conductors. Furthermore, we highlight the importance of recognising inherent challenges in transport mechanisms that involve multiple coordination changes along the pathway, attributing intrinsic minimum activation barriers to such scenarios. Notably, at certain lattice sizes, energies of different coordinations may become equivalent, leading to significantly lower barriers when a pathway involves a single coordination change. This insight enhances our comprehension of solid-state battery technology and facilitates the development of improved halide superionic conductors for future applications.

## Methods

### Computational

First-principles calculations were performed using the Vienna Ab Initio Simulation package (VASP)[36]. The generalised gradient approximation (GGA) exchange correlation with a Perdew-Burke-Ernzerhof (PBE) functional was adopted within the projector augmented wave (PAW) method[37]. The specific pseudopotentials that were used for each element are: Zr_sv, Cl, Li_sv, Na_pv, Cu_pv and Ag. For these calculations, a plane wave cut off of 520 eV was used with a $\Gamma$ centred $2 \times 2 \times 4$ k-point grid. Atom positions, cell volume and cell shape were allowed to relax until the forces acting on each atom reached less than 0.01 eV/Å. Some calculations were repeated using the meta-GGA exchange-correlation with the $r^2SCAN$ functional[38], increasing the plane wave cut-off to 600 eV.

All possible octahedral A-site orderings in the $P\bar{3}m1$ unit cell of $Li_2ZrCl_6$, $Na_2ZrCl_6$ and $Ag_2ZrCl_6$ were obtained using the Site Occupancy Disorder (SOD) code[39]. A-site orderings were considered for two separate Zr orderings along the c-axis: all Zr in a single layer (c = 0.5) or 2 Zr at c = 0 and 1 Zr at c = 0.5. Symmetrically distinct structures were relaxed using DFT, with the lowest energy taken as the ground state. The same strategy was employed for tetrahedral site occupancy in $Cu_2ZrCl_6$. Due to the large number of possible configurations (>1000) for $Cu_2ZrCl_6$, the Ewald summation method was employed to provide a basis for the low-energy structures based purely on electrostatics. The 240 lowest energy configurations were then relaxed among others. The $Cu_2ZrCl_6$ structure used in subsequent calculations was the lowest energy configuration where all Cu atoms had relaxed to tetrahedral positions.

Bader charge analysis was performed using the Bader charge analysis code[40] from the Henkelman group, using the VASP charge densities of the ground state calculations from the lowest energy $A_2ZrCl_6$ structures. Both the valence and core charge density were included in the calculation of the Bader charge.

Ab initio molecular dynamics (AIMD) calculations were performed to probe the transport properties in $A_2ZrCl_6$ and to calculate $A^+$ diffusivity. Plane-wave cutoffs were reduced to 400 eV to increase computational efficiency while using soft pseudopotentials. An NVT ensemble and a Nosé-Hoover thermostat were used to control the temperature of the simulations. $1 \times 1 \times 2$ supercells were used, which allowed the system lattice parameters to extend beyond 10 Å in each direction. K-point sampling was done using a $\Gamma$ centred $1 \times 1 \times 1$ mesh. All structures were equilibrated at their respective temperatures by performing a preliminary 10 ps simulation. AIMD calculations were performed using a 1 fs time step for 50 ps at their respective temperatures. Diffusivities of the A-site ions were calculated from their

mean square displacement (MSD) via equation (2):

$$D = \frac{MSD}{2dt} \quad (2)$$

Where d is the dimensionality of the system (3 in most cases) and t is the total time elapsed. MSD is calculated via equation (3):

$$MSD = \frac{1}{N} \sum_{i=1}^{N} |r_i(t + \Delta t) - r_i(t)|^2 \quad (3)$$

N is the number of ions for which the displacement is being calculated (number of $A^+$ ions in the supercell), $r_i$ is the position of the $i^{th}$ ion at time t, and $\Delta t$ is the time step.

Diffusivity and MSD values were calculated using the diffusion analyser module in pymatgen[41]. The lower bound of the temperature range for each system was selected such that an MSD value of >10 Å$^2$ for the A-site ion was achieved within the 50 ps simulation time to provide reliable estimates of the diffusion coefficient.

**Transition state searching.** Transition state searching (TSS) calculations were performed in the EON package[42] with input energetics from VASP. Unknown A-site activation barriers between the initial reactant state and a product state were located without previous knowledge of the final product states. To generate representative reactant state configurations with a range of A-site coordinations, supercells consisting of 2 $A_2ZrCl_6$ unit cells ($1 \times 1 \times 2$) were used. The positions of A-site ions in the supercell were initially thermalised with 10 ps of AIMD using a timestep of 1 fs under fixed cell conditions. At the end of the AIMD run, the positions of all ions were geometry optimised to a force tolerance of 0.01 eV/Å under fixed cell conditions.

For all TSS calculations, we utilised the DFT + D3 dispersion correction method proposed by Grimme[43] to treat Van der Waals interactions between layers. The absence of Van der Waals corrections was found to occasionally lead to artificial low energy barriers associated with the shearing of $ZrCl_6$ layers along the c-axis in the TSS approach. The PBE exchange-correlation functional was employed in combination with soft pseudopotentials for all elements, to minimise computational cost with an energy cutoff of 400 eV. Saddle point searches were then initiated from the reactant configuration by displacing all A-site ions in the cell. The magnitude of the displacement was based on a Gaussian distribution at 300 K with a standard deviation of 0.15. After the atoms were displaced, the transition state was located using the dimer method[44] under fixed volume conditions. The transition state geometries, containing a single negative mode, were converged to a force tolerance of 0.01 eV/Å using a conjugate gradient method. Once the transition state was found, the corresponding minima (reactant and product) were located by initiating minimisations along the negative and positive directions of the negative transition state mode. The force on all atoms was minimised to a tolerance of 0.01 eV/Å. The structure of the product state was compared to the reactant state to check that they were distinct minima. A product state was classified as a distinct minimum if the energy difference was > 0.02 eV and any atom had moved more than 0.2 Å.

**ACl Model.** For each ACl model system, a perfect, undistorted HCP structure was considered in which the Cl-Cl distance is equal to the a-lattice parameter $X_{Cl-Cl} = a$ and the ratio of the c to a lattice parameters was $\frac{c}{a} = \sqrt{\frac{8}{3}} = 1.667$. Configurations of A-site cations were considered for each material: A-site cations in octahedral sites ($A_{oct}$), A-site cations in tetrahedral sites ($A_{tet}$), A-site cations in trigonal planar sites within the ab-plane ($A_{trig,ab}$) and along the c-axis ($A_{trig,c}$) and A site cations in a linear configuration ($A_{lin}$).

The Cl-Cl distance ($X_{Cl-Cl}$) was chosen to describe the variation in the volume of the ACl lattice, as the Cl-Cl distance is invariant to the A-site coordination. An increase (decrease) in $X_{Cl-Cl}$ leads to an isotropic expansion (contraction) of the lattice. In the ideal HCP ACl lattice, $X_{Cl-Cl}$ is related to the A-Cl bond distance ($X_{A-Cl,i}$) via the relationships: $X_{A-Cl,oct} = \sqrt{2}/2X_{Cl-Cl}$, $X_{A-Cl,tet} = \sqrt{3/8}X_{Cl-Cl}$ and $X_{A-Cl,trig} = \sqrt{3}/3X_{Cl-Cl}$, for octahedral, tetrahedral and trigonal planar coordination, respectively.

## Experimental

**Synthesis of materials.** $ZrCl_4$ (99.5% Sigma Aldrich) and LiCl (99% Sigma Aldrich), NaCl (99% Sigma Aldrich), CuCl (99.999% Alfa Aesar) or AgCl (99.9% Thermo Scientific) were weighed in a 1:2 stoichiometric ratio and hand ground with a mortar and pestle inside an argon glovebox filled with $< 5$ ppm [$H_2O$] and [$O_2$]. The powders were then individually put into an air-tight zirconia jar with zirconia balls (20:1 weight ratio). The powders were milled using a planetary mill for 60 h at 400 rpm on alternating mode; there was a 5 min rest period between changing direction.

**X-ray diffraction.** Powder XRD patterns were collected inside a glovebox under argon using a Rigaku MiniFlex diffractometer with Cu K$\alpha$ radiation. No monochromator was used, leading to two wavelengths of radiation ($\lambda = 1.5406$ Å and $1.5444$ Å for K$\alpha_1$ and K$\alpha_2$, respectively). Measurements were conducted within the 10 to 90° $2\theta$ range at a rate of 0.1 degrees per minute. X-ray diffraction data was analysed using the Rietveld method using the GSAS-II software package[45].

**Electrochemical impedance spectroscopy.** Ionic conductivities of as-milled $A_2ZrCl_6$ samples were measured via AC impedance with a Biologic SP240 potentiostat. A pressure and atmosphere-controlled split cell was used to make SS|SE|SS (SS = stainless steel, SE = $A_2ZrCl_6$) symmetrical cells with a pellet diameter of 10 mm. Pellets are pressed in situ by the split cell. 600 MPa was applied for 5 min to allow the powder to densify prior to the measurement. An open circuit voltage with an amplitude of 50 mV was used with a frequency range of 7 MHz to 1 Hz. Subsequent data analysis was performed using the RelaxIS impedance analysis software. To obtain activation energies for the compounds, a climate chamber was used, and samples were allowed to equilibrate at each temperature for an hour prior to measurements being taken.

## Data availability

All data that support the findings in this work are available within the main article and supplementation information. The source data file: 494527_2_related_ms_9378315_sh61bw.xlsx is provided for the raw data for figures in the main text. Additional data relevant to this article are available from the corresponding author upon request. Source data are provided in this paper.

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

## Acknowledgements
K.B. and I.D.S. acknowledge the Imperial College Research Computing Service (10.14469/hpc/2232), and associated support services used during this work. This work was supported by the Royal Academy of Engineering (RCSRF/2021-1243). I.D.S. would like to acknowledge support for this work through a Royal Society of Edinburgh Research Award. We acknowledge Horizon Europe for project SEATBELT (101069726) and the Spanish Ministry for Science and Innovation for the project (TED 2021-129254B-C22). N.T.-R. would like to acknowledge the Faraday Institution FutureCat project (FIRG065) for the provision of some of the equipment used in this work.

## Author contributions
K.B. and I.D.S. conceived the idea for the study, which was planned with A.A. and S.J.S. K.B. performed the synthesis and electrochemical characterisation, with support and data interpretation from R.A, A.A., R.J. and I.D.S. Lab X-ray diffraction analysis was performed by K.B and S.L.M. with support and data interpretation from N.T.-R. and S.J.S. First-principles calculations were performed by K.B. and I.D.S. K.B. and I.D.S. wrote the manuscript with input from all authors.

## Competing interests
The authors declare no competing interests.
