## [Peer Review File · Nature Communications]

The importance of A-Site Cation Chemistry in Superionic Halide Solid ElectrolytesREVIEWER COMMENTS

Reviewer #1 (Remarks to the Author):

In this manuscript, A-site cation chemistry environment in A_2ZrCl_6 ($A=Li, Na, Cu, Ag$) system with $P-3m1$ space group was investigated by DFT calculation, and a series of A_2ZrCl_6 solid electrolytes were prepared by high-energy ball mill method. Among these electrolytes, the Cu_2ZrCl_6 and Ag_2ZrCl_6 were first prepared, exhibiting high conductivities of 10 and 4 $mS\ cm^{-1}$ at room temperature. A deep insight for activation energy was discussed based on transition state searching (TSS) method and ACI ($A=Li, Na, Cu, Ag$) HCP models. The employed ACI ($A=Li, Na, Cu, Ag$) HCP structures were used to study the relation between different A-site cation coordination number and cell energy, demonstrating that the ion transport activation energy depends on the nearest neighbor Cl-Cl distance. In this work, the authors strike out the microstructure of A-site cation (such as coordination number, kinetically resolved activation barriers, covalent) influences the conductivity of A_2ZrCl_6 , and the direction of conclusion is helpful for exploring new solid electrolytes. Overall, this paper can be considered to be accepted after the following issues are carefully considered and addressed.

1. I noticed that the "The refinement for Na_2ZrCl_6 utilised a minor $P-21/n$ phase to achieve the best fit", in Figure S3, there is only the Bragg position of $P-3m1$ space group, which is inconsistent. There are different XRD peaks on different A-site cations, the main peak index should be labeled. For the Rietveld refinement based on XRD, the Rwp is lacked in all refinement. And the χ^2 is higher, whether the presence of other phase lead to poor fit?
2. In Figure S4, I noticed that both A-site cations of Li and Na lead to the appearance of semicircles, but the cases of Cu and Ag without semicircles. What is the relationship between high conductivity and EIS semicircles? Please explain this phenomenon.
3. Some recent progresses of halide solid electrolytes also benefit from the structure and microstructure modulation, e.g. Adv. Funct. Mater. 2024, 2314044, DOI: 10.1002/adfm.202314044; ACS Nano 18, 5790-5804, 2024; Energy Storage Mater., 28, 37-46, 2020. These related works may be suitable to serve as references.
4. In Figure S6, for the AIMD calculation, Li_2ZrCl_6 , Na_2ZrCl_6 , and Cu_2ZrCl_6 all performed AIMD calculation at same temperature points, while Ag_2ZrCl_6 performed it at lower temperatures. AIMD simulation temperature of Ag_2ZrCl_6 is different from the temperatures of the other three systems. Is this the reason for the abnormal activation energy (0.09 eV)? At "Activation barriers and pathways of A^+ cations" section, the reason for the small activation energy of Ag^+ diffusion should be explained in detail. Moreover, Na_2ZrCl_6 discarded a data point at the lowest temperature when linear fitting. Please explain the reason for this.
5. According to that the Cl-Cl distance is equivalent to the lattice size parameter a . Is the change in Cl-Cl distance equivalent to the expansion or contraction of the lattice? The cell energy as a function of Cl-Cl distance is regarded as the Birch-Murnaghan equation of state. What is the difference between them? In the

unit cell, the A-Cl distance usually can better reflect the relationship between A ion migration and cell volume than the Cl-Cl distance. Please explain the reason of focusing on the Cl-Cl distance rather than the A-Cl distance.

6. In the AIMD calculation section, the supercells of A_2ZrCl_6 were used, the cell parameters and structure model (e.g. POSCAR file) should be provided.

Reviewer #2 (Remarks to the Author):

In this manuscript, the authors rationalized why the ionic conductivity of Cu_2ZrCl_6 (10 mS cm^{-1}) is higher than other A_2ZrCl_6 ($A = \text{Li, Na, Ag}$). They explained that the role of A-site chemistry is important in fast ion conduction by analyses using Rietveld refinement for XRD, DFT, AIMD, TSS calculations. Specifically, it reads that the main conclusion is that for specific mobile ion with proper ionic size can render energy landscape during ion transport through intermediate states become remarkably flattened. It is meaningful that the experimental results could be interpreted theoretically. However, while the authors emphasized the occupancy of Cu ions in the tetrahedral sites, which is distinct from others, I hardly found its intuitive correlation with the aforementioned energy barrier variations. Moreover, there is a lack of close inferences in analysis of experiments. Several points that could help improve this manuscript are as follows.

1. Rietveld table of P21/n phase of Na_2ZrCl_6 should be provided. Bragg index of the P21/n phase in Figure S3 should also be shown.
2. R_p , R_{wp} , R_{exp} , R_{bragg} values for each Rietveld profile should be provided.
3. Please discuss why only Cu exists in the tetrahedral site among four elements despite the similar ionic radius to Li.
4. Please compare the R & Chi values for the result of Rietveld fitting of Cu_2ZrCl_6 with Cu in tetrahedral sites versus octahedral site.

Reviewer #3 (Remarks to the Author):

This article explores the model system A_2ZrCl_6 ($A = \text{Li, Na, Cu, Ag}$) to understand the fundamental role that A-site chemistry plays on fast ion transport. Having synthesised A_2ZrCl_6 for the first time, they reveal exceptional room temperature ionic conductivities in Cu_2ZrCl_6 and Ag_2ZrCl_6 of 1×10^{-2} and $4 \times 10^{-3} \text{ S cm}^{-1}$, respectively. It proposes that the energy and number of transition states play pivotal roles in ion transport. The models in this article provides a deeper understanding into the optimisation and design criteria for halide superionic conductors. It can be published in the Nature Communications after the minor corrections. Detailed comments are as follows:

1. The accurate and valid Rietveld refinement should be based on the data in wider diffraction angle instead of just $0-90^\circ$ shown in the Figure S3. As we know, the XRD pattern for Li_2ZrCl_6 can not be assigned with the pattern of Li_3YCl_6 since there is an overall peak shift and there is no standard CIF file for Li_2ZrCl_6

- itself, consequently, what CIF file, the standard crystal structure file, do you conduct the Rietveld refinement with? Besides, R_{wp} and R_p during the Rietveld refinement should be provided to measure the refinement accuracy.
2. There are two wavelength values for the XRD measurement, why?
 3. The different atoms in Figure 1 should be described to make the crystal structure clear.
 4. In Page 3, what does it mean for "that is at $c = 0$ "? There is no description about the c parameter.
 5. In Page 4, the word Na in the sentence "Na has the largest activation barrier of 0.58 eV" should be Na_2ZrCl_6 .
 6. As the capacitance in Figure S4 has been measured via fitting in Page 4, please provide the equivalent circuit for the EIS results in the supporting information.
 7. Please provide some details or basic introduction about Bader charge analysis in the Experimental section.
 8. The Year field in ref. 23, 26, 34 should be corrected.

Reviewer comments are shown in black. Author responses in blue. Manuscript/SI changes in brown below, and highlighted in the manuscript and SI.

RESPONSE TO REVIEWERS' COMMENTS

Reviewer #1 (Remarks to the Author):

In this manuscript, A-site cation chemistry environment in A_2ZrCl_6 (A=Li, Na, Cu, Ag) system with $P-3m1$ space group was investigated by DFT calculation, and a series of A_2ZrCl_6 solid electrolytes were prepared by high-energy ball mill method. Among these electrolytes, the Cu_2ZrCl_6 and Ag_2ZrCl_6 were first prepared, exhibiting high conductivities of 10 and 4 mS cm^{-1} at room temperature. A deep insight for activation energy was discussed based on transition state searching (TSS) method and ACI (A=Li, Na, Cu, Ag) HCP models. The employed ACI (A=Li, Na, Cu, Ag) HCP structures were used to study the relation between different A-site cation coordination number and cell energy, demonstrating that the ion transport activate energy depends on the nearest neighbor Cl-Cl distance. In this work, the authors strike out the microstructure of A-site cation (such as coordination number, kinetically resolved activation barriers, covalent) influences the conductivity of A_2ZrCl_6 , and the direction of conclusion is helpful for exploring new solid electrolytes. Overall, this paper can be considered to be accepted after the following issues are carefully considered and addressed.

We thank the reviewer for the encouraging words about the content of the manuscript. We have addressed the related comments below:

1. I noticed that the “The refinement for Na_2ZrCl_6 utilised a minor $P2_1/n$ phase to achieve the best fit”, in Figure S3, there is only the Bragg position of $P-3m1$ space group, which is inconsistent. There are different XRD peaks on different A-site cations, the main peak index should be labeled. For the Rietveld refinement based on XRD, the Rwp is lacked in all refinement. And the c2 is higher, whether the presence of other phase lead to poor fit?

The Na_2ZrCl_6 data shows a minor $P2_1/n$ phase which is required to provide the best fitting. However, the inclusion of the $P2_1/n$ phase likely led to a higher χ^2 value compared to the others due to the additional and overlapping peaks. The lack of Bragg positions for the minor $P2_1/n$ phase was an oversight and have now been included. Refinements have been revised to ensure all the data are accurate. Figure S3 has been updated (below) to show the Bragg indices for both phases as well as all relevant R values. An additional comment has been added to the caption of Figure S3:

A secondary $P2_1/n$ phase was required to adequately fit the Na_2ZrCl_6 sample. The presence of this secondary phase led to a slightly higher χ^2 value (9.70) compared to the Li_2ZrCl_6 , Ag_2ZrCl_6 and Cu_2ZrCl_6 refinements containing a single $P-3m1$ phase where $\chi^2 < 6$. The broadness of the peaks, associated with reduced crystallite sizes as a result of the harsh milling process, increases the difficulty of refining X-ray diffraction data. To avoid over fitting of broad peaks lead to non-physical values, some isotropic thermal displacement parameters were fixed.

2. In Figure S4, I noticed that both A-site cations of Li and Na lead to the appearance of semicircles, but the cases of Cu and Ag without semicircles. What is the relationship between high conductivity and EIS semicircles? Please explain this phenomenon.

An explanation of why high conductivity requires increasingly higher frequencies to resolve the associated processes has been included in Supplementary note 2:

Semicircles are not observed for Cu_2ZrCl_6 or Ag_2ZrCl_6 . The time constant of a conduction process is related to the resistance and capacitance of the system:

$$\tau = RC$$

The frequency f at which a process is observed in EIS is related to τ via the angular frequency ω :

$$\omega = 2\pi f$$

$$\omega = 1/\tau$$

As conductivity is inversely proportional to resistance, we can see that very large conductivities require increasingly higher frequencies (>7 MHz in this case) to resolve the conduction process.

Two additional references have been added to the main text to support the use of the x intercept as a value of the resistance, R (Tanaka, Y. *et al* *Angewandte Chemie*. 2023, Warner *et al*, *Journal of Solid State Chemistry*. 1992).

3. Some recent progresses of halide solid electrolytes also benefit from the structure and microstructure modulation, e.g. Adv. Funct. Mater. 2024, 2314044, DOI: 10.1002/adfm.202314044; ACS Nano 18, 5790-5804, 2024; Energy Storage Mater., 28, 37-46, 2020. These related works may be suitable to serve as references.

We thank the reviewer for these suggestions. We have added Energy Storage Mater., 28, 37-46, 2020 as a reference as we feel this is relevant for the current manuscript. We amended the below sentence including the reference:

Chlorides and fluorides show the best performance regarding electrochemical stability and are lighter than bromides and iodides, however they display lower ionic conductivities. [Energy Storage Mater., 28, 37-46, 2020]

4. In Figure S6, for the AIMD calculation, Li_2ZrCl_6 , Na_2ZrCl_6 , and Cu_2ZrCl_6 all performed AIMD calculation at same temperature points, while Ag_2ZrCl_6 performed it at lower temperatures. AIMD simulation temperature of Ag_2ZrCl_6 is different from the temperatures of the other three systems. Is this the reason for the abnormal activation energy (0.09 eV)? At “Activation barriers and pathways of A^+ cations” section, the reason for the small activation energy of Ag^+ diffusion should be explained in detail. Moreover, Na_2ZrCl_6 discarded a data point at the lowest temperature when linear fitting. Please explain the reason for this.

We thank the reviewer for this important point. Due to the computational cost of AIMD simulations, the use of high temperatures is typically required to allow for ionic transport processes to be observed on the relatively short (50 ps) timescales of the simulations. The diffusion coefficient D in Figure S6 is extracted from a linear fit of the mean-square-displacement (MSD) against simulation time. In order to get reliable values of D , the lower temperature limit for each material was selected so that the MSD for the A-site cation was at least 10 \AA^2 . The activation energy for Ag_2ZrCl_6 was initially fit over a lower temperature range as the low activation energy for Ag_2ZrCl_6 meant that MSD values greater than 10 \AA^2 could be achieved, even at 350 K. In order to make the data more comparable to the other materials systems, additional high temperature AIMD runs have been included for the Ag_2ZrCl_6 system in Figure S6 at 650 K and 700 K (see below), and the fitting of the activation energy has now been performed over the full range.

The following sentence have been added to the methods section:

The lower bound of the temperature range for each system was selected such that a MSD value of $> 10 \text{ \AA}^2$ for the A-site ion was achieved within the 50 ps simulation time to provide reliable estimates of the diffusion coefficient.

For the Li_2ZrCl_6 and Cu_2ZrCl_6 systems, we have also increased the number of data points to 8 within the same temperature range to improve the reliability of the fit. The new activation energies after fitting are shown in the updated Figure S6 and are updated in the main text, which agree closely with the previous results.

For the Na_2ZrCl_6 system, previous molecular dynamic studies have shown that there is a change in the activation energy for conduction above 600 K. We therefore restricted our fitting of the activation energy to the high temperature points above 600 K. On closer analysis, the MSD of the data point at 600 K was also below 10 \AA^2 due to the high activation energy for the Na_2ZrCl_6 system, which results in a high uncertainty in the value of D . We have therefore removed this point from Figure S6 and focused the fitting of the energy to the high temperature regime above 600 K, analogous to previous studies. The raw MSD vs time data for all systems have been added to the additional data folder.

The following sentences and additional references have been added to supplementary note 3:

For the Na_2ZrCl_6 system, a discontinuity in the diffusion coefficient was previously observed below 600 K, (Wu, E. A. et al, Nature Communications (2021), Sebti, E. et al, Journal of Materials Chemistry (2022)). The activation energy for Na conduction in the ‘high temperature’ regime was therefore extracted from a linear fit to data points above 700 K.

5. According to that the Cl-Cl distance is equivalent to the lattice size parameter a . Is the change in Cl-Cl distance equivalent to the expansion or contraction of the lattice? The cell energy as a function of Cl-Cl distance is regarded as the Birch-Murnaghan equation of state. What is the difference between them? In the unit cell, the A-Cl distance usually can better reflect the relationship between A ion migration and cell volume than the Cl-Cl distance. Please explain the reason of focusing on the Cl-Cl distance rather than the A-Cl distance.

We thank the reviewer for this point. Calculations in this work were performed to compare the energetics of different A-site cations (Li, Na, Cu and Ag) in octahedral, tetrahedral and trigonal planar sites within an ideal hexagonal close packed ACl lattice. The close-packed Cl lattice makes up the framework of the HCP ACl structure, so the size of the octahedral, tetrahedral and trigonal planar sites is dictated by the Cl-Cl ($X_{\text{Cl-Cl}}$) distance at each volume. The A-Cl bond distance, however, changes depending on whether the A-cation is in an octahedral, tetrahedral or trigonal planar site. The variation in the A-Cl bond distance with A-site coordination hinders a direct comparison between different structures, which is why the Cl-Cl distance was chosen, as it is invariant to the A-site coordination. Previous works have used similar metrics based solely on the anion lattice, such as the ‘anion volume’, in reference 32. $X_{\text{Cl-Cl}}$ can be related to the A-Cl ($X_{\text{A-Cl},i}$) distances for octahedral, tetrahedral and trigonal planar sites via the relationships: $X_{\text{A-Cl},\text{oct}} = \sqrt{2}/2 X_{\text{Cl-Cl}}$, $X_{\text{A-Cl},\text{tet}} = \sqrt{3}/8 X_{\text{Cl-Cl}}$ and $X_{\text{A-Cl},\text{trig}} = \sqrt{3}/3 X_{\text{Cl-Cl}}$.

As highlighted by the reviewer, and mentioned in the paper, for the HCP lattice, the Cl-Cl ($X_{\text{Cl-Cl}}$) distance is equal to the a-lattice parameter. The c-lattice parameter can also be described in terms of $X_{\text{Cl-Cl}}$ via $c = (8/3)^{1/2} X_{\text{Cl-Cl}}$. An increase in the Cl-Cl distance therefore results in an isotropic expansion of the lattice volume. The variation in the cell energy as a function of the lattice volume could be fit with the Birch-Murnaghan, or equivalent, equation of state expression to extract quantities such as the bulk modulus. This fitting was, however, not performed in the current work as the mechanical properties of the materials were not the focus of this study.

We have included the sentences below into Methods Section of the main text to clarify these points. We believe that some of the issues identified above may have come from the occasional use of the term 'Cl-Cl bond distance', so we have changed this term wherever present to 'Cl-Cl distance' to make it clearer.

The Cl-Cl distance ($X_{\text{Cl-Cl}}$) was chosen to describe the variation in the volume of the ACI lattice, as the Cl-Cl distance is invariant to the A-site coordination. An increase (decrease) in $X_{\text{Cl-Cl}}$ leads to an isotropic expansion (contraction) of the lattice. In the ideal HCP ACI lattice, $X_{\text{Cl-Cl}}$ is related to the A-Cl bond distance ($X_{\text{A-Cl},i}$) via the relationships: $X_{\text{A-Cl},\text{oct}} = \sqrt{2}/2 X_{\text{Cl-Cl}}$, $X_{\text{A-Cl},\text{tet}} = \sqrt{3/8} X_{\text{Cl-Cl}}$ and $X_{\text{A-Cl},\text{trig}} = \sqrt{3}/3 X_{\text{Cl-Cl}}$, for octahedral, tetrahedral and trigonal planar coordination, respectively.

6. In the AIMD calculation section, the supercells of A_2ZrCl_6 were used, the cell parameters and structure model (e.g. POSCAR file) should be provided.

As requested by the reviewer, we have included a zip folder containing all the relevant structure and input files for all the calculations used in this work.

Reviewer #2 (Remarks to the Author):

In this manuscript, the authors rationalized why the ionic conductivity of Cu_2ZrCl_6 (10 mS cm^{-1}) is higher than other A_2ZrCl_6 ($\text{A} = \text{Li, Na, Ag}$). They explained that the role of A-site chemistry is important in fast ion conduction by analyses using Rietveld refinement for XRD, DFT, AIMD, TSS calculations. Specifically, it reads that the main conclusion is that for specific mobile ion with proper ionic size can render energy landscape during ion transport through intermediate states become remarkably flattened. It is meaningful that the experimental results could be interpreted theoretically. However, while the authors emphasized the occupancy of Cu ions in the tetrahedral sites, which is distinct from others, I hardly found its intuitive correlation with the aforementioned energy barrier variations. Moreover, there is a lack of close inferences in analysis of experiments. Several points that could help improve this manuscript are as follows.

We thank the reviewer for their comments and discussion of the current work. The manuscript and SI has been updated to address their considerations.

1. Rietveld table of $P2_1/n$ phase of Na_2ZrCl_6 should be provided. Bragg index of the $P2_1/n$ phase in Figure S3 should also be shown.

We refer to our responses to reviewer 1 in which the diffraction data has been updated to include all the relevant Bragg indices. A table of refinement results for the $P2_1/n$ phase has also been included as supplementary table 5:

Table S5. Rietveld refinement result from the room-temperature X-ray powder diffraction data of the as-milled Na₂ZrCl₆. The space group is *P2₁/n*. The refined lattice parameters are a = 6.665 Å, b = 7.091 Å, c = 9.810 Å. β = 92.224°.

Atoms	x	y	z	Occ.	site	Sym.	U _{iso}
Zr1	½	½	½	1	2b	2	0.036(4)
Na1	0.530(4)	0.085(3)	0.287(24)	1	4e	4	0.021(-)
Cl1	0.266(27)	0.203(31)	0.941(20)	1	4e	4	0.059(11)
Cl2	0.107(26)	0.947(4)	0.238(21)	1	4e	4	0.073(9)
Cl3	0.652(27)	0.773(25)	0.440(19)	1	4e	4	0.013(8)

2. Rp, Rwp, Rexp, Rbragg values for each Rietveld profile should be provided.

We thank the reviewer for highlighting this point. As discussed above in the response to reviewer 1, point 1, Figure S3 has been updated to include all R values associated with the revised refinements.

3. Please discuss why only Cu exists in the tetrahedral site among four elements despite the similar ionic radius to Li.

We have added the following explanation in the main text as to why Cu(I) exists in tetrahedral rather than octahedral sites:

Li⁺ has been found in both octahedral and tetrahedral configuration in halide-based materials, depending on the size of the lattice and presence of neighbouring vacant sites. Although Li⁺ and Cu⁺ have similar ionic radii, additional mixing between the 3d¹⁰ valence orbitals in Cu⁺ with the higher energy 4s and 4p states leads to an additional stabilisation of the tetrahedral configuration, as shown in previous work [Inorganic Chemistry 31, 1758–1762 (1992), Angewandte Chemie, 12275–12279 (2017)]

4. Please compare the R & Chi values for the result of Rietveld fitting of Cu₂ZrCl₆ with Cu in tetrahedral sites versus octahedral site.

A refinement was conducted where the positions of Cu atoms were located on the octahedral sites, as was found to be the case for the other species (Li, Na and Ag). Upon refinement, the atomic positions would deviate drastically from the octahedral sites. Fixing the Cu atoms in these octahedral sites resulted in a much poorer fit ($\chi^2 = 17.9$, $R_{wp} = 7.85$) than the refinement conducted starting with Cu in tetrahedral sites. The following line has been added to the main text to discuss this point.

A refinement was conducted for Cu₂ZrCl₆ where the Cu⁺ species were located on octahedral sites. This refinement led to a poor fit with χ^2 and R_{wp} values of 17.90 and 7.85 respectively).

Reviewer #3 (Remarks to the Author):

This article explores the model system A₂ZrCl₆ (A = Li, Na, Cu, Ag) to understand the fundamental role that A-site chemistry plays on fast ion transport. Having synthesised A₂ZrCl₆ for the first time, they reveal exceptional room temperature ionic conductivities in Cu₂ZrCl₆ and Ag₂ZrCl₆ of 1 × 10⁻² and 4 × 10⁻³ S cm⁻¹, respectively. It proposes that the energy and number of transition states play pivotal roles in ion transport. The models in this article provides a deeper understanding into the optimisation and design criteria for halide superionic conductors. It can be published in the Nature Communications after the minor corrections. Detailed comments are as follows:

We would like to thank the reviewer for their kind words on the content and conclusions of the current work.

1. The accurate and valid Rietveld refinement should be based on the data in wider diffraction angle instead of just 0-90° shown in the Figure S3. As we know, the XRD pattern for Li_2ZrCl_6 cannot be assigned with the pattern of Li_3YCl_6 since there is an overall peak shift and there is no standard CIF file for Li_2ZrCl_6 itself, consequently, what CIF file, the standard crystal structure file, do you conduct the Rietveld refinement with? Besides, Rwp and Rp during the Rietveld refinement should be provided to measure the refinement accuracy.

The mechanochemical milling process leads to an emphasised broadening of the peaks towards higher angles ($>90^\circ 2\theta$). As a result of this, we proceeded with a refinement of the data at 10 – 90° 2 θ . While Li_2ZrCl_6 has no direct CIF file recorded on the ICSD, tabular powder neutron diffraction refinement data is included in the following reference, which was used for the refinement in this work: 'Wang, K. et al. A cost-effective and humidity-tolerant chloride solid electrolyte for lithium batteries. *en. Nature Communications* **12**, 4410 (2021).'

The Rwp and Rp values among others have been added to Figure S3 as seen is our responses to reviewer 1.

2. There are two wavelength values for the XRD measurement, why?

We would like to highlight that the diffractometer utilised lacks a monochromator, hence the samples are irradiated with both Cu $K_{\alpha 1}$ and $K_{\alpha 2}$ radiation. This has been emphasised the experimental section:

Powder XRD patterns were collected inside a glovebox under argon using a Rigaku MiniFlex diffractometer with Cu K_{α} radiation. No monochromator was used leading to two wavelengths of radiation ($\lambda = 1.5406 \text{ \AA}$ and 1.5444 \AA for $K_{\alpha 1}$ and $K_{\alpha 2}$ respectively).

3. The different atoms in Figure 1 should be described to make the crystal structure clear.

We have added an atom key into Figure 1 to make the crystal structures easier to visualise:

4. In Page 3, what does it mean for “that is at $c = 0$ ”? There is no description about the c parameter.

We thank the reviewer for highlighting the confusion this comment might cause. “ $c = 0$ ” refers to the atoms occupying the space shared by unit cells in the c -direction. We note that due to periodic boundary conditions this information is redundant, and the remark has since been removed from the manuscript.

5. In Page 4, the word Na in the sentence “Na has the largest activation barrier of 0.58 eV” should be Na_2ZrCl_6 .

We thank the reviewer for spotting this error. The main text has since been amended.

6. As the capacitance in Figure S4 has been measured via fitting in Page 4, please provide the equivalent circuit for the EIS results in the supporting information.

Equivalent circuit models were used to fit the data of Li_2ZrCl_6 and Na_2ZrCl_6 only. The x-intercept was used to obtain resistance values for calculating the conductivities of Cu_2ZrCl_6 and Ag_2ZrCl_6 . Figure S4 has been updated to include the equivalent circuit models and component values used for Li_2ZrCl_6 and Na_2ZrCl_6 :

The following text been added to the figure caption for Figure S4.

The equivalent circuit models consisting of a parallel resistor/constant phase element and a series Warburg element that were used to fit the resistance and capacitance of the Li₂ZrCl₆ and Na₂ZrCl₆ systems are shown in (a) and (b). The resistance for the Cu₂ZrCl₆ and Ag₂ZrCl₆ systems was taken from the x-intercept.

7. Please provide some details or basic introduction about Bader charge analysis in the Experimental section.

We have added a description of the parameters used in the Bader charge calculations to the experimental section of the main text:

Bader charge analysis was performed using the Bader charge analysis code (Henkelman, G., Arnaldsson, A. & Jónsson, H. Computational Materials Science (2006)) from the Henkelman group, using the VASP charge densities of the ground state calculations from the lowest energy A₂ZrCl₆ structures. Both the valence and core charge density were included in the calculation of the Bader charge.

Some additional background into Bader charge analysis was also added in in Supplementary note 3:

Supplementary note 3: Bader charge analysis

Bader charge analysis is a way of determining the distribution of electronic charge within a molecule or solid by defining the contributions from individual atoms. This is achieved by finding where the electron density gradient is zero, representing the areas of bonding between atoms. These areas of electron density can be used to find the charge associated with each atom. These charges can be compared to that of the species formal charges to see the bonding character between species. Bader charge analysis was performed using the Bader charge analysis code.[36]

8. The Year field in ref. 23, 26, 34 should be corrected.

The year for each of these references has been added.

REVIEWERS' COMMENTS

Reviewer #1 (Remarks to the Author):

This revised version can be accepted in its current form.

Reviewer #2 (Remarks to the Author):

The authors have addressed the comments by the reviewers.
The quality of the manuscript has been improved.

Reviewer #3 (Remarks to the Author):

the authors have revised the manuscript according to the comments in detail. I suggest it can be accepted for publication